# Experimental assessment of the lidar polarizing sensitivity

L. Belegante<sup>1</sup>, J. A. Bravo-Aranda<sup>2,3</sup>, V. Freudenthaler<sup>4</sup>, D. Nicolae<sup>1</sup>, A. Nemuc<sup>1</sup>, L. Alados-Arboledas<sup>2,3</sup>, A. Amodeo<sup>5</sup>, G. Pappalardo<sup>5</sup>, G. D'Amico<sup>5</sup>, R. Engelmann<sup>6</sup>, H. Baars<sup>6</sup>, U. Wandinger<sup>6</sup>, A. Papayannis<sup>7</sup>, P. Kokkalis<sup>7</sup>, and S. N. Pereira<sup>8</sup>

<sup>1</sup>National Institute of Research and Development for Optoelectronics, 409 Atomistilor Str, Magurele, Romania <sup>2</sup>Andalusian Institute for Earth System Research, Granada, Spain

<sup>3</sup>Department of Applied Physcis, Unversity of Granada, Granada, Spain

<sup>4</sup>Meteorological Institute, Ludwig-Maximilians-Universitat, Theresienstr. 37, 80333 Munich, Germany

<sup>5</sup>Istituto di Metodologie per l'Analisi Ambientale CNR-IMAA, C.da S. Loja, Tito Scalo, Potenza 85050, Italy

<sup>6</sup>Leibniz Institute for Tropospheric Research, Permoserstr. 15, 04318 Leipzig, Germany

<sup>7</sup>National Technical University of Athens (NTUA), Physics Department, Heroon Polytechniou 9, 15780 Zografou, Athens, Greece

<sup>8</sup>Evora Geophysics Center, Rua Romao Ramalho 59, 7000, Evora, Portugal

Correspondence to: J. A. Bravo-Aranda (jabravo@ugr.es)

**Abstract.** Particle depolarization ratio retrieved from lidar measurements are commonly used for aerosol typing studies, microphysical inversion, or mass concentration retrievals. The particle depolarization ratio is one of the primary parameters that can differentiate several major aerosol components, but only if the measurements are accurate enough. The uncertainties related to the retrieval of particle depolarization ratios are the main factor in determining the accuracy of the derived parameters in

- such studies. This paper presents an extended analysis of different depolarization calibration procedures, in order to reduce the related uncertainties. The calibration procedures are specific to each lidar system of the European Aerosol Research Lidar Network - EARLINET with polarising capabilities. The results illustrate a significant improvement of the depolarization lidar products for all the selected lidar stations. The calibrated volume and particle depolarization profiles at 532 nm show values that agree with the theory for all selected atmospheric constituents (several aerosol species, ice particles and molecules in the paragraphic free regions).
- aerosol free regions).

#### 1 Introduction

Uncertainties related to the influence of anthropogenic activities on the Earth's energy budget and climate change have lead to a real interest regarding the aerosols direct and indirect radiative effects. Measurements of vertically resolved aerosol optical properties (as the one performed by lidar systems) try to reduce these uncertainties. These systems are laser-based instruments

able to provide quantitative information on aerosol layering and their properties (Measures et al., 1992). The principle is based on the detection of backscattered light that results from the interaction of the emitted laser light with the atmospheric constituents. Fig.1 shows the main components of a lidar system with polarising capabilities.. The emitted laser light is oriented towards the atmosphere by means of the emission optics. After the emitted light interacted with atmospheric constituents, the backscattered light is collected by a telescope and directed to the wavelength separation unit (WSU- named also the receiving

optics unit for this study), Polarizing Beam Splitter (PBS) and photomultipliers (PMTs). The receiving optics (mirrors, lenses and dichroic filters), the PBS, and the PMTs will be treated as distinct units, since the effects of each unit alters the depolarization profiles from different perspectives. The collected laser light contains information about to the optical properties of the atmospheric components, and consequently to their size, shape and composition. Methods to retrieve these properties from elastic backscatter, Raman, multi-wavelength and depolarization lidars are already described in details in the literature (Fernald

- et al., 1972; Klett et al., 1981, 1985; Kovalev and Eichinger, 2004). According to their application, lidar systems have different configurations, channel combinations and geometries. For atmospheric studies, the configuration of a lidar system narrows down to several types and optical layouts. A major breakthrough in atmospheric studies is the development of global lidar networks, able to provide systematic lidar dataflow with a large temporal and spatial coverage (Earlinet , 2014a). The European
- Aerosol Research Lidar Network EARLINET data (Pappalardo et al., 2012, 2014) is relevant for climatology, regional and large scale models but also for special events such as Saharan dust outbreaks, transport of smoke plumes or volcanic ash over Europe (Earlinet , 2014b, d, e), (Papayannis et al., 2008; Ansmann et al., 2009; Ansmann and Bosenberg, 2003; Nicolae et al., 2013; Mona et al., 2012). The multi-wavelength depolarization Raman lidar systems used in EARLINET (3β+2α+δ lidar systems Mona et al., 2012), are capable to provide an extended set of optical parameters for aerosol characterization, by assuring
- the quality of the products through internal data quality procedures. For depolarization studies, most of the lidar systems are designed to measure with two channels (parallel and cross). Recent atmospheric studies using remote sensing data have been dedicated to aerosol typing, microphysical inversion and aerosol mass concentration retrievals. Since all relevant parameters are shape dependent (Hervo et al., 2012; Hogan et al., 2012; Gross et al., 2015), the depolarization products obtained from lidar measurements proved to be essential, giving the opportunity to distinguish between rather spherical particles with low
- depolarization ratios, and non-spherical particles with higher depolarization ratios (Gasteiger et al., 2014). Still, most questions are related to the accuracy of the lidar products used for these studies (lidar ratio, particle linear depolarization ratio, Ångström exponent). Lidar measurements of particle linear depolarization ratio are often used to discriminate between low depolarizing (e. g. local aerosol) and high depolarizing (dust) aerosols, or liquid and ice clouds, requiring only a relative measure of these parameters. At the present state, the uncertainties of these products are high and any aerosol classification based on relative
- lidar depolarization profiles is challenging. For aerosol typing and mass concentration studies, absolute values of particle linear depolarization ratio are needed. According to (Petzold et al., 2010; Gross et al., 2013; Burton et al., 2012), the particle linear depolarization values characterizing several aerosol species (or mixtures of aerosols) ranges around similar values: for aged dust, the particle depolarization value at 532nm ranges from 0.21 to 0.29 and for pure dust from 0.30 to 0.39 with a lidar ratio (LR) overlapping from 40 to 60 sr for aged dust and 35 to 63 sr for pure dust respectively. The same issue emerges
- when discriminating between biomass burning aerosol mixed with mineral dust and industrial pollution aerosol, with values around 0.1 to 0.2 (LR 36-90 sr) for the first and 0.04 to 0.1 (LR 35-70 sr) for the second. Therefore, in order to discriminate between different type of particles, the uncertainty of these products has to be significantly improved, e.g. by reducing and / or correcting for instrumental errors. Currently, most lidar depolarization studies are performed assuming ideal lidar optics with no influence on the final depolarization products (volume and particle linear depolarization ratio). Recent studies showed
- that even small deviations from the ideal assumptions can lead to large uncertainties of the retrieved depolarization products.

Typically, the main source of uncertainty does not come from the detected signal (noise), but from systematic errors in the optical setup of the lidar systems (Freudenthaler et al., 2009; Freudenthaler, 2015; Alvarez et al., 2006; Snels et al., 2009; Biele et al., 2000). One of the most efficient method in measuring the absolute value of depolarization parameters is by implementing hardware depolarization calibration methods. This study aims to present several techniques developed to calibrate the

- lidar depolarization channels. Moreover, methods for assessing the influence of lidar optics on depolarization products (i.e., the assessment of the receiving optics diattenuation parameter and the rotation of the plane of polarization of the laser around the light propagation axis with respect to the PBS) in order to reduce the corresponding uncertainties (Mattis et al., 2009) are discussed. The paper presents an extended analysis of different depolarization calibration procedures specific to each lidar setup (or cluster of similar systems) in EARLINET. The analysis is focused on the two channels setup, designed to measure
- the backscattered signal components polarised parallel and perpendicular (cross) with respect to the linear polarization of the emitted light, but can be further developed to be applied for many optical layouts. The first part of the paper describes the theoretical background, architecture and methodology used for depolarization calibration procedures, including a broad description of the available calibration procedures, but also new techniques for performing the calibration of lidar depolarization channels. New methods to retrieve the influence of different optical modules on depolarization products are also presented and
- discussed. Methods to assess and correct the rotation of the plane of polarization of the laser around the light propagation axis  $(\alpha)$  are also introduced and discussed. Section 2 describes the theoretical background based on the Mueller-Stokes formalism used as the basis for the entire study. The methodology is given in section 3. The second part of the paper shows results of calibrated and not-calibrated lidar depolarization profiles, several case studies from different lidar instruments in EARLINET, discussions and conclusions. Volume and particle linear depolarization ratios are presented, emphasizing the added value of
- calibrated depolarization channels, especially when quantitative information is envisaged. The results show the impact of the studies (calibration of the depolarization channels and influence of the optics) on primary lidar products and implicitly on aerosol typing results. Section 4 shows the results and discussions and the conclusions are given in section 5. The experimental approach of the paper is designed to present how depolarization calibration procedures are implemented. Most of the available literature is focused on the theoretical perspective of the topic and practical issues usually remain an open topic. A hands on
- approach for the assessment of the lidar polarization sensitivity is most welcomed since most of these calibration techniques require comprehensive practical descriptions that must be treated thoroughly.

#### 2 Theoretical background

The Mueller-Stokes formalism (Chipman, 2009; Ossikovski et al., 2010; Lu and Chipman, 1996) describing the lidar system setup (shown in Fig.1) can be summarized by the following equation (Freudenthaler, 2015):

$$I_{S} = \eta_{S} \mathbf{M}_{S} \mathbf{R}(y) \mathbf{M}_{0}(\gamma, D_{0}) \cdot \mathbf{F}(a) \mathbf{M}_{E}(\beta) I_{L}(\alpha, a_{L})$$
(1)

where bold italic fonts are used for the Stokes vectors, bold for the Mueller matrices and italic for the scalar variables.  $I_L(\alpha, a_L)$  is the Stokes vector of the light emitted by the laser,  $M_E$  is the Mueller matrix of the emission block optics, F represents the Mueller matrix of the atmospheric scattering volume in backscattering direction, a is the polarization parameter of the atmospheric volume described below in more detail,  $M_0$  is the receiving optics matrix characterized by the receiving optics

- diattenuation parameter  $D_0$ ,  $\mathbf{R}(\mathbf{y})$  is the rotation matrix, y describes the optical setup type (see Fig 2.a-b).  $\mathbf{M}_S$  stands for both parts of the polarizing beam splitter, i.e. the transmitted (subscript T) and reflected (subscript R) channels, including additional polarizing elements after the polarizing splitter. The incident plane of the polarizing beam splitter is taken as the reference plane for all rotation angles around the optical axis.  $\eta$  represents the calibration factor including only the electronic amplification and the optical diattenuation of the two polarizing channels and  $I_S$  is the Stokes vector for the two detected channels (i.e.  $I_R$  and
- $I_T$ ) (see also Fig 1) where the Stokes vector describes the polarisation state of the measured channels (either transmitted or reflected).  $\alpha$  is the rotation of the plane of polarization of the laser around the propagation axis,  $a_L$  is the polarization parameter of the light beam leaving the laser,  $\beta$  is the rotation of the emitter optics around the propagation axis,  $\gamma$  is the rotation of the receiver optics around the propagation axis. In order to have a complete characterization of the lidar optics, the contribution of all latter parameters must be accounted. The technological solutions for mounting the receiving optics and the PBS are
- based on high precision optical mounts for all the considered lidar setups. These implementations assure high accuracy and minimization of any rotation misalignment of the optics. The analyzed EARLINET lidars have the  $\gamma$  and  $\beta$  angles lower than 0.5° as indicated by (Bravo-Aranda et al., 2015). Since the larger uncertainties are expected for  $D_O$  and  $\alpha$ , we neglect the effect of  $\gamma$  and  $\beta$  angles on these lidar systems.

A significant simplification comes from the "v" component of the emitted Stokes vector (i,q,u,v) (Chipman, 2009), (Lu and
Chipman, 1996), (Freudenthaler, 2015), (Ossikovski et al., 2010). By neglecting this component, we assume that the emitting optics does not have retardation effects. This simplification can be performed once the *α* parameter is corrected and the *β* and *γ* angles are negligible.

The polarizing beam splitter cross talk can also be neglected since all lidar systems described in this study are equipped with additional polarization filters placed after the PBS on both transmitted and reflected channels, aiming to minimize this effect in an efficient way. The polarization parameter of the light beam leaving the laser  $a_L$  will be neglected since for this study, the

in an efficient way. The polarization parameter of the light beam leaving the laser  $a_L$  will be neglected since for this study, the polarization of the emitted light will be considered 1. This does not apply for all lidar systems, but for simplification reasons we will consider instruments equipped with ideal light sources. In this study we consider only the *gain ratio*  $\eta^*$ , the *rotation of the plane of polarization of the laser around the propagation axis*  $\alpha$  and the *diattenuation parameter of the receiver optics* D<sub>0</sub>. Freudenthaler (2015) describes in details, all the terms and the variables present in Eq. (1). According to that, the Mueller

matrix describing the atmospheric backscatter is:

$$\mathbf{F} = \begin{pmatrix} F_{11} & 0 & 0 & 0 \\ 0 & F_{22} & 0 & 0 \\ 0 & 0 & -F_{22} & 0 \\ 0 & 0 & 0 & F_{44} \end{pmatrix} = \\ F_{11} \begin{pmatrix} 1 & 0 & 0 & 0 \\ 0 & a & 0 & 0 \\ 0 & 0 & -a & 0 \\ 0 & 0 & 0 & 1-2a \end{pmatrix}$$
(2)

5 
$$a = \frac{F_{22}}{F_{11}}$$
 (3)

Consequently, the linear depolarisation ratio of the atmospheric scattering volume (
$$\delta$$
) can be defined as

$$\delta = \frac{F_{11} - F_{22}}{F_{11} + F_{22}} = \frac{1 - a}{1 + a} \Rightarrow a = \frac{1 - \delta}{1 + \delta}$$
(4)

All optical elements  $\mathbf{M}_O$  can be described by Mueller matrices of diattenuators  $\mathbf{M}_D$  with retardation  $\mathbf{M}_{ret}$ :

$$\mathbf{M}_{O} = \mathbf{M}_{D}\mathbf{M}_{ret} = T_{O} \begin{pmatrix} 1 & D_{O} & 0 & 0 \\ D_{O} & 1 & 0 & 0 \\ 0 & 0 & Z_{O} & 0 \\ 0 & 0 & 0 & Z_{O} \end{pmatrix} \cdot \begin{pmatrix} 1 & 0 & 0 & 0 \\ 0 & 1 & 0 & 0 \\ 0 & 0 & c_{O} & s_{O} \\ 0 & 0 & -s_{O} & c_{O} \end{pmatrix} = T_{O} \begin{pmatrix} 1 & D_{O} & 0 & 0 \\ D_{O} & 1 & 0 & 0 \\ 0 & 0 & Z_{OCO} & Z_{OSO} \\ 0 & 0 & -Z_{OSO} & Z_{OCO} \end{pmatrix}$$
(5)

with

10

$$T_{O} = \frac{T_{O}^{p} + T_{O}^{s}}{2}, D_{O} = \frac{T_{O}^{p} - T_{O}^{s}}{T_{O}^{p} + T_{O}^{s}}, Z_{O} = \frac{2\sqrt{T_{O}^{p}T_{O}^{s}}}{T_{O}^{p} + T_{O}^{s}} = \sqrt{1 - D_{O}^{2}},$$

$$c_{O} = \cos\Delta_{O}, s_{O} = \sin\Delta_{O}, \Delta_{O} = \varphi_{O}^{p} - \varphi_{O}^{s}$$
(6)

where  $\Delta_O$  is the differential phase shift of the p and s light components and  $T_O^p$ ,  $T_O^s$  are the optics intensity transmission for 15 parallel (p) and cross (s) linearly polarised light with respect to the plane of incidence of the PBS. We don't investigate the effects of the emitter optics described by the Mueller matrix  $\mathbf{M}_E$  in this study, since at least two of the considered lidars send

the laser radiation in the atmosphere without using any optic (MUSA and MULIS). For lidar systems that use emission optics to send the laser radiation in the atmosphere, further investigations are needed to fully characterize the effects of  $\mathbf{M}_E$  on the depolarization products (Bravo-Aranda et al., 2015). We also assume that the light emitted to the atmosphere,  $\mathbf{I}_E$ , is linearly polarised with an angle  $\alpha$  with respect to the reference plane, which reduces Eq. (1) to

5 
$$I_{S} = \eta_{S} \mathbf{M}_{S} \mathbf{R}(\mathbf{y}) \mathbf{M}_{O}(\gamma, D_{O}) \mathbf{F}(a) I_{L}(\alpha)$$
 (7)

The calibrated signal ratio including cross talk and alignment errors  $\delta^*$  can be calculated from the measured signals using the calibration factor  $\eta$ :

$$\delta^* = \frac{1}{\eta} \cdot \frac{I_R}{I_T} \tag{8}$$

In order to determine the calibration factor η, we use the calibration methods described further on in the paper. The general
formula for the retrieved light intensity is described in (Freudenthaler, 2015), Sect. 4. The detected light intensity for the p and c components can be described by

$$\frac{I_{S}(\mathbf{y},\varepsilon,\gamma,a,\beta,\alpha)}{\eta_{S}T_{S}T_{O}F_{11}T_{E}I_{L}} = (1+\mathbf{y}D_{S}D_{O}\mathbf{c}_{2\gamma+2\varepsilon})i_{E} - \mathbf{y}D_{S}Z_{O}\mathbf{s}_{O}\mathbf{s}_{2\gamma+2\varepsilon}v_{E} + \\
+aD_{O}\left(\mathbf{c}_{2\gamma}q_{E} - \mathbf{s}_{2\gamma}u_{E}\right) + \\
+ayD_{S}\left(c_{2\varepsilon}q_{E} + s_{2\varepsilon}u_{E}\right) - \\
-ayD_{S}s_{2\gamma+2\varepsilon}\left(W_{O}\left(s_{2\gamma}q_{E} + c_{2\gamma}u_{E}\right) - 2Z_{O}s_{O}v_{E}\right) \tag{9}$$

For the total signal (Fig 2.c), the detected light intensity can be described by

$$\frac{I_{tot}(\mathbf{y},\varepsilon,\gamma,a,\beta,\alpha)}{\eta_t T_t T_O F_{11} T_E I_L} = i_E + \mathbf{a} D_O \left( \mathbf{c}_{2\gamma} q_E - \mathbf{s}_{2\gamma} u_E \right)$$
(10)

20 where  $c_{...} = cos(...)$ ,  $s_{...} = sin(...)$ ,  $\varepsilon$  is the error angle of the  $\Delta 90^{\circ}$  calibration setup and *i*, *q*, *u*, *v* are the Stokes components of the emitted light.

$$W_O = 1 - Z_O c_O, \quad D_S = \frac{T_S^p - T_S^s}{T_S^p + T_S^s}$$
(11)

For horizontal or vertical linear polarised laser and without rotational misalignment of the emission optics, receiver optics, laser and of the calibrator, i.e.

25 
$$u_E = v_E = 0, i_E = q_E = +1 \land \gamma = \varepsilon = 0 \land a_L = 1$$
 (12)

we have

15

$$\frac{I_S(y,0,0,a,0,0)}{\eta_S T_S T_O F_{11} T_E I_L} = (1 + y D_S D_O) + a \{ D_O + y D_S \}$$
(13)

(15)

and with a cleaned polarising beam splitter (additional polarization filters placed after the PBS to minimize the cross talk)

$$D_R = -1, D_T = +1 \Rightarrow D_S = \pm 1 \tag{14}$$

resulting

$$\frac{I_S(y,0,0,a,0,0)}{\eta_S T_S T_O F_{11} T_E I_L} = 
= (1 \pm y D_O) + a (D_O \pm y) =
= (1 \pm y D_O) + ya (y D_O \pm 1) = 
= (1 \pm y D_O) \pm ya (1 \pm y D_O) = 
= (1 \pm y D_O) (1 \pm ya)$$

we then have

$$\frac{I_R}{I_T} = \frac{\eta_R T_R}{\eta_T T_T} \frac{(1 - yD_O)(1 - ya)}{(1 + yD_O)(1 + ya)}$$
(16)

where  $\eta_R$  and  $\eta_T$  are the electronic amplification of individual transmitted/reflected channels.  $T_R$  and  $T_T$  are the transmission and reflectance for un-polarised light passing through the PBS.

With Eqs. (4) and (16)

$$y = +1 \Rightarrow \delta = \frac{1}{\eta} \frac{I_R}{I_T} \frac{1+D_O}{1-D_O} = \frac{1}{\eta^*} \frac{I_R}{I_T}$$
 (17)

for

$$\mathbf{y} = -1 \Rightarrow \delta = \eta \frac{I_T}{I_R} \frac{1 + D_O}{1 - D_O} = \eta^* \frac{I_T}{I_R} \tag{18}$$

with

$$\eta = \frac{\eta_R T_R}{\eta_T T_T} \tag{19}$$

the calibration factor that only takes into account detection efficiencies and optics transmission after the PBS. For more general or other special cases see (Freudenthaler, 2015).

## 3 Methodology

## 3.1 Determination of the gain ratio $\eta^*$ : calibration procedures

The calibration of depolarization channels is specific to each lidar system, but the basic principles are similar for most of the 25 instruments. The calibration of the depolarization channels consists of assessing the gain ratio  $\eta^*$ . This parameter includes the calibration factor but can also include the cross talk from optics before the polarising beam splitter and system alignment errors.

In order to determine the gain ratio  $\eta^*$ , the first solution is to use the "0° calibration" or the "atmospheric calibration". Using this calibration, the contribution of the system to the final lidar depolarization products is assessed by using an aerosol-free range in the lidar signal (Freudenthaler et al., 2009), an altitude where only the molecular contribution has to be considered. In a such atmospheric region, the total volume linear depolarisation ratio can be approximated by the well known value of the air molecule linear depolarisation ratio (Behrendt and Nakamura, 2002). Usually this calibration (normalization), applied to the depolarization profile does not take into account all the effects that have to be corrected in the depolarization profile (i.e., crosstalk coefficient, diattenuation effects, phase shift, etc.), resulting in erroneous depolarization values especially in highly depolarizing aerosol (e.g. ice crystals) in the assumed clean range which can easily lead to large errors in the final depolarization products (Freudenthaler et al., 2009; Freudenthaler, 2015).

A better and more reliable solution to calibrate the depolarisation measurements is represented by the "45° calibration". This calibration implies a 45° rotation of the depolarization analyzer (PBS and the PMTs) with respect to the polarization plane of the laser, in order to equalize the light intensity in the cross and parallel channels. When comparing the calibration signals, the difference between the transmitted and reflected channels is given by the contribution of optics and electronics

- in the lidar receiving unit, e.g. diattenuation effects of optical components in the receiving unit, detection efficiencies of the PMTs, different optics in front of the detectors like the presence of any neutral density filters, and different gains of the pre-amplifiers and amplifiers for each channel. The 45° rotation between the input (laser) polarization plane and the polarising beam splitter axis can be technically implemented by physical (mechanical) rotation of the whole depolarization analyzer or by the rotation of an additional half wave plate (HWP) optic (Fig 3.b-c). A further solution can be the use of a polarising filter
- at  $45^{\circ}$  with respect to the incident plane of the polarising beam splitter. The implementation of this methods will be further described in this study.

(Freudenthaler, 2015) describes these calibration techniques in a quite general context. In this study we consider only the rotation of the emitted laser polarisation and the diattenuation of the receiving optics. The gain ratio is determined using

$$\eta^*(y,x) = \frac{I_R(y,x)}{I_T(y,x)}$$
(20)

with  $x = \pm 1$

The main source of uncertainty involved in this kind of calibration is represented by the accuracy in determining the rotation of  $45^{\circ}$ . The less is this accuracy the large is the errors in estimating the calibration constant. A better solution is to use two subsequent measurements performed by rotating the depolarization analyzer at  $\pm 45^{\circ}$  with respect to the default measuring position. This calibration is called the " $\pm 45^{\circ}$  calibration". The calibration constant is determined by using the geometric mean

of the two  $\pm 45^{\circ}$  measurements. The two measurements are designed to compensate each other even for cases where the  $45^{\circ}$  rotation uncertainty is significant (Freudenthaler et al., 2009). A more general solution is to use two subsequent measurements performed by rotating the depolarization analyzer with an exact 90° difference between each other. This calibration method is called the " $\Delta 90^{\circ}$  calibration" and the output is similar with the one from the  $\pm 45^{\circ}$  calibration. The " $\pm 45^{\circ}$  calibration" can

be considered a particular case of the " $\Delta 90^{\circ}$  rotation calibration" since the only constrain of this calibration is the  $90^{\circ}$  angle between the two measurements.

Technically, the " $\Delta 90^{\circ}$  calibration" can be implemented by using a mechanical rotator (holder), which rotates the optical components at fixed  $\Delta 90^{\circ}$  angles. This calibrator will be further called the " $\Delta 90^{\circ}$  mechanical rotation calibrator". A similar

- approach (same output) can be considered if we use a HWP for accurately rotating the emitted or collected light at  $\Delta 90^{\circ}$ . The advantage is that while the mechanical rotator can only be placed in the reception unit (in front of the receiving optics or in front of the PBS), the HWP module can be also placed at the emission, in front and after the emission optics. This calibrator will be further called the " $\Delta 90^{\circ}$  HWP calibrator". A third approach of the " $\Delta 90^{\circ}$  calibration" is the use of an additional linear polarizer that can be rotated at fixed  $\Delta 90^{\circ}$  angles. In this case, the  $\Delta 90^{\circ}$  rotation will be replaced by the additional linear
- polarizer. According to its position in the optical chain (in front of the telescope or the PBS in the receiving unit) the calibration can account for all lidar optics placed after the polarizer (e.g. receiving optics, PBS, PMT). Further on, this calibrator will be called the " $\Delta 90^{\circ}$  polarizer rotation calibrator". In order to perform the latter calibration, the 'zero' position of the optical module (polariser or HWP) in respect to the relative position of the PBS must be determined and corrected for. For this, the  $\Delta 90^{\circ}$  rotation calibration requires an extra measurement set to assess the offset angle between the calibrator and the zero
- position of the PBS.

For exemplification, the error angle of the calibration setup ( $\varepsilon$ ) must be estimated to allow a reliable measurement of the calibration constant using the  $\Delta 90^{\circ}$  polarizer calibration. In order to determine  $\varepsilon$ , a set of two relative  $\pm 45^{\circ}$  measurements is required. The polarizer is placed in a random position relative to the polarization plane of the receiving optics. Two measurements will be performed with the polarizer rotated precisely at  $\pm 45^{\circ}$  from the  $\varepsilon$  angle.

20 In cases when the  $\pm 45^{\circ}$  calibration factor can be described by

$$\frac{\eta^*(y,x,\varepsilon)}{\eta} = f(y,\dots)\frac{1+Kxs_{2\varepsilon}}{1-Kxs_{2\varepsilon}}$$
(21)

with  $K \le (1 - a)$  parameter taking into account instrument contributions except x and  $\varepsilon$ . It is possible to determine the calibration rotation  $\varepsilon$  from

$$Y(\varepsilon,K) = \frac{\eta_{pol}^*(y,+45^\circ,\varepsilon,K) - \eta_{pol}^*(y,-45^\circ,\varepsilon,K)}{\eta_{pol}^*(y,+45^\circ,\varepsilon,K) + \eta_{pol}^*(y,-45^\circ,\varepsilon,K)} = \frac{2Ks_{2\varepsilon}}{1+K^2s_{2\varepsilon}^2}$$
(22)

$$\varepsilon = \frac{1}{2} \arcsin\left[\frac{1}{K} \tan\left(\frac{\arcsin\left(Y\left(\varepsilon, K\right)\right)}{2}\right)\right]$$
(23)

the formulas can be applied to:

- polarizer before the PBS:

$$30 \quad \frac{\eta^*(y,x,\varepsilon)}{\eta} = \frac{1 + xys_{2\varepsilon}}{1 - xys_{2\varepsilon}} \tag{24}$$