# Peer review of "Experimental assessment of the lidar polarizing sensitivity"

_Atmospheric Measurement Techniques, 2015_

## Referee Comment (RC1) · Anonymous Referee #1 · 1 Mar 2016

The depolarization ratio is an important property to characterize different aerosol types. An exact measurement of this quantity is therefore an important issue. The manuscript describes a method to assess and to correct for the diattenuation of receiving optics and the rotation angle between the laser and the receiver. It is shown that these corrections lead to a significantly better result for the volume depolarization ratio (fig 9). The lidar community will profit from these techniques. Therefore I recommend it for publication, although there are some mayor points that have to be improved:

Firstly I dare that the figures do not fulfill the standard of the journal. A professional plot program should be used.

Secondly There is no clear hint, that the shown 6 example cases are corrected with all the effort described in this paper. The diattenuation of the receiving optics (D0) is

given for 4 out of 6 lidar systems; the laser rotation alpha is given for the Bucharest system only. If you assess experimentally the lidar polarization sensitivity you have to do it for every system that you are presenting in the paper. It is an AMT publication and you use 6 out of 14 figures just to give examples for different atmospheric conditions without showing the improvement due to your correction scheme or assessing the polarization sensitivity. It is already known that exact depolarization measurements are important, so please do not show only the measurements, but the improvement due to your corrections (with and without calibration).

Thirdly For all figures (fig 4,5,6), if alpha does not exceed 10-15° in real situations (as you say), figures have to be limited to this range. You may show the whole range at the beginning (fig 4a), but later on, only real situations are of interest for the reader and not the theoretical maxima and minima of your simulation. You lose a lot of relevant information by using these large scales.

I strongly recommend improving these 3 points before publishing the manuscript.

The following comments should be considered and answered before publication. They are meant to improve the quality of the paper and consider related aspects.

Comments: (p-page; l-line; eq-equation)

p4 l19-22 How do you separate the influence of the angle $\alpha$ and a possible circular component resulting in elliptical polarized emitted light?

p5 l16 2 lidar systems without emitter optics out of 6 investigated systems is no reason to neglect this part.

You should include setup c (total and cross) in your paper in order to describe the different depolarization setups. Add comments on setup c in eq 17 and 18. The manuscript seems incomplete in this point.

p9/10 It would be helpful to add a sketch to explain your estimation of the calibration error $\varepsilon$. Please check (p9 l22) $K \leq (1-a)$; for aerosol free region (a=1), K would be

$\leq 0$. From eq 29 to eq 30 it is a simple transformation, which would imply $\varepsilon 2 = \varepsilon/\varepsilon 1$. But how do you define your $\varepsilon 2$? And p10 l6 is $\eta$ really dependent on the atmospheric depolarization or should it be $\eta^*$?

The wavelength dependence of several parameters and calibration methods should be mentioned.

The simulations are a central part of the paper, assessing the sensitivity of different parameters. A description of the simulation and all the values used for the Bucharest lidar system are missing.

Minor/Technical comments:

p2 l16 "recent atmospheric studies" -> Citation

p2 l32 "uncertainty": the before mentioned values vary due to natural variability and uncertainties are not yet mentioned

p3 l16 in the introduction it is not necessary to mention "$(\alpha)$"

p5 eq 6 $\varphi$ is not explained

p7 eq 15 It should be mentioned that $y=\pm 1$, otherwise the transformation is not correct. It would be nice to include this information ($y=\pm 1$) in figure 2, too.

p8 l28 "depolarization analyzer" or polarization analyzer?

p9 l10/11 "According to ... PBS, PMT)." This sentence is valid for every calibration method, not only for the linear polarizer.

p12l4 "polarization Raman lidar" not "depolarization Raman lidar"

Fig 4, very interesting results, but to emphasis the message, split fig 4b for high and low depolarization ratios or use a logarithmic scale, otherwise your threshold of $3°$ seems somehow arbitrary. You say (p13l27/28) that alpha will not exceed 10-15°, so fig 4c should be plotted only till 20° not 50°. p13l27 "the dependence between the retrieved

calibrated signal ratio and alpha is more significant" please include relative errors in the text to demonstrate this. Small comments: figure description 4b is plotted (0°:10°) Why do you use 0.4 in figure 4c and 0.35 in fig 4a and 4b?

eq 35 after removing the polarizer, there should be no $\eta\_pol^*$

p15, chap 3.4.3, a suggestion: add a sub headline "Analytical correction" analogous to "Correction of alpha in front of the PBS"

fig 6a+6b is identically to fig 4a+4b; $\delta^*$ $(\alpha)$ in exactly the same range is shown.

p17l4-7 To derive the diattenuation of the receiving optics it is not necessary to have 2 different calibration methods (pol and rot), but to have 2 different places in the optical setup. To determine D0 it is necessary to calibrate before and after M0. So I would suggest calling it "before" and "after" instead of "pol" and "rot".

---

## Referee Comment (RC2) · Anonymous Referee #2 · 23 Mar 2016

Overall Comments: The topic of lidar depolarization measurements is important and relevant to many science investigations to better understand the optical properties of aerosol and clouds. It is also agreed that the calibration of lidar systems is important and must be done carefully and accurately. Therefore, new methods or approaches to make these calibration or to assess them is very relevant. However, it is not clear in this paper that has been achieved.

Overall this manuscript is too long, complicated, and lacks clear logical flow that is not satisfactory to be published. It needs a lot of streamlining and careful thought into the organization to highlight the key motivation and results, which appears to be the improvement to the polarization calibration approach and procedures. The paper leverages heavily from concurrent manuscripts submitted that likely go into specific details about the theory of the calibrations and assessment of measurement accuracy

of the depolarization measurements. This paper does not need to redo what is covered by those manuscripts but should focus on those elements that are covered in the experimental assessment of the sensitivity and accuracy. The paper addresses many existing systems with a variety of approaches and designs that are difficult to keep track of in the paper. Even, a simple table of the objective (calibrations), approach to calibrations, and the pro and con statements for each. Also, it would be good if the specific new and novel information was highlighted in the paper. Most of this seems to be similar or slight variants to techniques in the literature already so this needs to be clearly articulated in the paper. The results are lacking in significance. No final assessment of the accuracy of the measurements is provided, but only changes if the systems are grossly misaligned or not properly calibrated to start. The current conclusion are not substantiated in the manuscript that one technique is better than the others. It not clearly stated that there is new or unique methods of this paper that is not already in the literature.

If this has provided new methods for the different lidar systems that changed and improved the data quality then this should be addressed for each of those systems to allow the data users more direct knowledge of those changes. If this is the case, the first half of this paper should be condensed significantly to address only those specific improvements. The conclusions and the corrections that are made should be the focus not additional information about the theory that are ignored in these assessments. Are substantial conclusions reached? There is a statement of which calibration techniques are better considering the current EARLINET configurations in the conclusion section yet these conclusions are not supported by data. However, this could be done in a much more succinct way; first highlight the methods, deficiencies, and then provide the solutions and the quantitative assessment of improvement. Quite frequently, the motivation for the derivation was not explicitly stated or is lost in the details, an example of being the diattenuation assessment. It was not clear why this is needed until later in the paper it was discussed that current techniques to calibrate ignore upstream optics. The title should include that this is for the EARLINET lidar systems as it is clearly targeted

for that purpose. The methodology of the depolarization calibration seem reasonable but the uniqueness was not presented and the assessment of them does not provide specific results like system accuracy for each method. The value of this to the instrument teams might be useful but could be covered under a technical document. The value to the data users is not clear as no assessment of accuracy of the measurements were provided other than some blanket final values of accuracy without discussion.

Major revisions should include: 1. Provide clear distinction of the intended audience, is this intended to be for EARLINET lidars or polarization lidars in general. Likely this is intended for the EARLINET lidars discussed, if so then this should be stated up front and in the title. Is this intended for the instrument teams to provide calibration methods that are better than currently implemented or to the data users? If it is intended for data users then the paper must include the specific errors and changes in the data resulting from the calibrations for the various systems. 2. Provide clear description of the objectives. Is this to demonstrate one technique of calibration is better than others or this to demonstrate that all the techniques are satisfactory if done correctly? 3. If required, provide a much more logical and simplified approach to the theory. In the end, this paper only addressed a couple aspects but carries a complicated methodology that is reduced significantly in the end. 4. The flow of the paper needs to be improved with the end goal provided before all the theory and methodology is developed. This will be improved by understanding the objectives and what is required by the various systems. 5. A clear description of the various techniques and instruments is required. 6. The conclusions need to clearly state what is new and different that has been addressed from the previous literature and must be supported by the measurements.

Additional General Comments Another important aspect is that the calibration of these systems likely (most do) change over time and it is not clear how stable they are over time. It seems that this should and would be a major consideration to reach a confident accuracy assessment. For example, detectors can have temperature effects that change the gain ratio, any changes to the laser like alignments can effect these calibrations. These concerns were not addressed in the manuscript. . In addition, it seems that any retardation differences in the two components is not carried through in the analysis. In particular the ellipticity in the system is a real issue that would need to be at least addressed. This is a case where the complex framework described somewhat includes this concern but it is quickly dropped from the test and verification.

If the system is grossly calibrated then large errors can occur as demonstrated by these examples, but it does not provide an assessment of the accuacy of the methods or the various systems in EARLINET. This seems to be a significant part of the motivation to determine the best methods of calibration and assess the accuracy. Discussions of pros and cons are made but this is not a rigorous assessment. The results appear to be if you calibrate the system properly then all the methods are good. This raises the overall question of what improvements in the measurement were made by this effort. In the end, can this paper provide an assessment of the accuracy of the various lidar systems (at least due to calibrations)? And can this be incorporated into the current and future datasets? What is the sensitivity and accuracy of the polarizing lidars in EARLINET?

Specific Comments:

Abstract The theory for the various particles likely have a range of values that can be realized in the atmosphere. Suggest that the authors state that these fall within a range of values that are generally accepted. For example, I am would trust the theory for the ice particles optical properties over the measurements.

Introduction The introduction is nearly two pages long and is only one paragraph. Suggest organizing this to separate the main points. Page 1 Line 18: interacts rather than interacted. Page 2 Line 2 and Line 29: It is recommended to not reference unpublished manuscripts unless it refers to where additional information may be found (for example Freudenthaler 2015). (this is also the case in other locations of the manuscript). Page 2 Line 11: suggest changing models to assessment of models. Page 2 Line 13:

the three beta + 2 alpha + depol (symbols are not defined). Page 2 Line 24: suggest change to ... these products are high for EARLINET lidars and .... Page 2 Line 28: definitions of aged and pure either need references or this could be combined into dust and dust mixtures with the full range of values stated. The reader will get caught at trying to assess the aerosol typing in the paper rather than focus on the objective to better calibrate the depolarization measurements. (relates to next comment). Page 2 Lines 20-31: this discussion might be relevant to the motivation for accurate measurements but since this deals with the depolarization measurements it would be much better to shorten this part of the paper to focus on a few cases as simple examples highlighting the need for better depolarization measurements (maybe just discuss the spectral depolarization studies of dust for example). This argument is not really strong anyway as there can be a range of values just due to mixtures of aerosol types. Since this paper is focused on the depolarization measurement, the discussion on the lidar ratio also seems a bit distracting to the main objective. Page 2 Lines 32: For these sentences it would be best to 1) state the current estimated errors, 2) state why it needs improvement (depol measurements) 3) state the goals that you are trying to achieve. This would be much more direct and simple than what is currently in the manuscript. Page 2 Line 34: suggest the following change: Currently, most EARLINET lidar depolarization studies are performed... or otherwise provide references suggesting most lidar studies are performed assuming this to be the case. Page 2 Line 35: provide references for these studies. Page 3 Line 6-7: The statement in parentheses is the only assessment made so I suggest removing the parenthesis. Page 3 Line 13 - 14: It is not clear what the difference between new techniques and new methods mean in these sentences? Seems to be a repeat of the same thing so you might change the wording. Page 3 Line 15: Again it seems to state the same thing once again. These three sentences should be combined into a simple statement without repeating the same underlying point. Page 3 Line 13 -26: This section needs to be written more clearly and it can be much more succinct. If the overall comments above are addressed then this section will flow much better as well. Page 3 Line 20 Envisaged? Suggest using

necessary or required instead. Page 3 Line 25: Not sure what sentence is intended to mean specifically and needs clarification. Does this mean that this manuscript is needed for proper operation and analysis of the EARLINET lidar systems?

Theoretical Background Page 4 Line 11: What does aL represent? It is stated that this is the polarization parameter without description. Page 4 Line 20-22: This might be a big assumption if not measured. It states that the ellipticity of the outgoing laser is negligible. Surprisingly, it is stated later that some of the systems have no optics beyond the laser output which means that the laser is assumed to produced perfect linear polarization beams. It is stated that that this simplification can be performed once the rotation angles are determined but it is not clear how why is the case. Page 4 Line 25: Here it is assumed that the laser has perfect linearly polarized light. This is a bit confusing when first read in the paper. Again it seems that there should be some verification of the assumption for the systems described. Page 4 Line 29: It would be good to rewrite equation on Page 3 Line 30 with all these assumptions before providing all the matrix definitions. In the end, this paper addresses the rotation angle of the laser outgoing pulse to the plane of the analyzer and the optical and electronic gain of the receiver assuming that the laser polarization is purely linear and the separation is pure without cross-talk. All the other details are then assumed to be perfect. This is a bit frustrating to go through all the math and details and then just ignore them. It would be much better to ignore them at the beginning and then derive the equations if this is the intent of the paper. At this point in the paper it would be good to state why these are the two key factors in the analysis and this is all that is needed to be analyzed. It seems that we have setup a complicated formulism and now have reduced this to just these two parameters that are the typical parameters being assessed for most polarization lidar systems. Page 5 Line 9: It would be good to expand on the retardation matrix. First it would be good to give a published reference and second it should provide some context on what this matrix provides. For example, does it assume that the spatial variations along the beam profile provide the same retardation? In other words could there be differences along the beam profile in the optics due to different surfaces?

There are cases where this might be important. Page 5 Line 10: Are the subscripts necessary? This makes it more difficult to follow and makes the equations with much smaller fonts than probably necessary. Page 5 Line 14: Please define variables when first introduced and stick to one convention. Page 5 Line 15 through top of Page 6: This is not a good reason to ignore what is being transmitted. I suggest either the authors provide measured values or provide a better rationale. Page 6 Line 6: Here we are including the cross-talk and earlier it was stated that we were assuming that this was zero. Is this a different cross-talk? What is the new parameter delta*? To be honest, it is hard to follow the discussion at this point and forward. Page 7 Line 12-15: A whole new series of equations are presented with reference to an unpublished paper without complete definition of the variables.

Currently this entire section needs to be revised for the following reasons: There are new variables not defined, the motivation for all the equations are lacking, the complexity of the equations are not required for this paper as they are simplified at end. Why not start with the assumption first and then the equations become very simple. It also seems that the retardation in the system is ignored but this is not stated at this point, so the reader is left wondering its value. In summary, I do not think that it is possible to fully follow the derivation presented here. In fact, it is not clear what eta* represents and this is the main discussion of the next section.

Methodology As noted above, it is not clear what the variable eta* represents so the following is an attempt to add further review without complete knowledge of the definitions. Page 7 Line 25 Finally, eta* is defined as including the cross-talk of the receiver optics and system alignment errors. By cross talk I assume that this includes rotation of the polarization only and not any induced ellipticity. Needs to be more clear, later in the paper I think that we are talking about the diattenuation and the cross talk of the polarizer is still assumed negligible. Page 8 Line 2: This is a region assumed to have low aerosol loading likely estimated from the lidar backscatter profile. Suggest not using aerosol-free region. Page 8 Line 4: Remove 'a' and replace with 'an' - In such

an atmospheric region... Page 8 Line 5-9: This basically states that this calibration approach is not very reliable. Why not just state that this approach is not reliable and move on thus removing most of this paragraph. In fact, there are probably references stating that this is the case. On line 1 it is stated that this is a solution, rather this is an approach that is not accurate typically. If it not accurate then this is not a solution. Page 8 Line 12: change 'implies' to 'implements'. Page 8 Line 15: This sentence can be made much shorter and simple without the need for all the examples. Also please check the wording as one usually compares the ratio not the difference in the signals. Page 8 Lines 17-21: Are these needed for this paper? I would just clearly state the methods implemented later in the paper. As stated below, this really needs a simple table with names and labels to keep this separated and straight. Page 8, Line 25: The variable 'x' is not defined. Suggest using the same parameters as noted earlier. Page 8 Line 27: Suggest rewording this sentence to more clear. The errors in the calibration are driven by the accuracy of the rotation mechanism. The two angles can potentially provide better accuracy at the expense of doing more calibration procedures or it takes longer, in particular if the angle of the of the laser and the receive analyzer are not matched well. This brings up another question as to the stability of the calibrations and how often are they required. This might be bigger error than those due to the calibration biases. Page 8 and 9: The discussion of the different calibration approaches are finally presented and a table (simple one) would summarize this nicely. I suggest that a table be made to highlight the pros and cons. Also it seems that the SNR will come into play for some of these calibration or it will take more time to implement the calibrations. It seems appropriate to consider these trades. In other words, some of the signals might be weak depending on the angle of the polarizer if I understand this correctly. Page 9, line 16: suggest changing exemplification to example. Page 10: The methodology is presented in very complicated manner and with additional effort could be easily presented to have a reader understand the basic principle. There are likely several methods to set this angle to at least a reasonable accuracy very quickly. For example, it is the polarization rotation angle could be simply estimated by looking

at relatively clean and stable atmospheric regions and minimizing the cross polarized signal to at least provide a reasonable first iteration. In addition, one would look at minimizing the difference at complementary angles (+/-) from the assumed angle and iterating (this assumes that there is no ellipticity in the laser beam or retardation effect in the receiver). The end goal is to have an accurate alignment of the polarization angle. This could be provided in a table with the different methods and the pros and cons to summarize this in a more logical and succinct form. Page 11 Lines 2-6: It is stated for the first time that the diattenuation products can be assessed. This should be clearly stated at the beginning of the paper that these methods are to provide more than just the required calibration constants needed for depolarization measurements. The confusion comes because it is not stated that there are some EARLINET lidars that can or do not measure the full optical path and these need to be assumed and therefore new calibration techniques presented address these issues.

Determination of the gain ratio eta*: Experimental solutions As noted earlier, these approaches using the rotation of the transmitted beam or the received light is typical of those found in the literature. It would be good to reference those systems. For example, the NASA CALIOP instrument implements a similar approach by inserting a depolarization optic in the receiver to calibrate the lidar gain ratio very successfully (reference below). The concern here is that it is not clear what is new or what makes these techniques unique beyond what has been reported. Early on it is stated that there is a lack of literature to understand the proper way to provide polarization calibration and I agree that there are pitfalls without careful design, however this paper needs to go beyond the current literature. (Winker, David M., William H. Hunt, and Matthew J. McGill. "Initial performance assessment of CALIOP." Geophysical Research Letters 34.19 (2007)).

Page 11 Line 25: By inserting and removing an optic in the receiver for calibration, care must be taken to ensure that it does not change the alignment. Many detectors are position sensitive and changes in Again, there is always pros and cons to a technique

and a table might be the best option to highlight the trades and this should highlight the new and unique methods/techniques. Page 12 Line 7: It is noted that some polarization optics at the focus of the beam (field stop) could present issues. It depends on the speed of the focus and also the polarizer properties and without the specific details this would be a concern. Page 12 Line 9: Suggest deleting either latter or further on (not both).

Assessment of the diattenuation parameter Do Is this required for the calibrations of the depolarization? If not then there needs to be some motivation of why this is needed in this paper. I am confused on why this needs to be separated in the analysis. It is stated that simulations show that this can be a significant effect on the depolarization products. In what way? SNR or calibration? If the diattenuation is not accounted for in the full receiver chain for the calibrations then yes this can be a significant issue. Maybe this paper is getting at some weaknesses in the current calibration approaches for each system but not providing those to the reader clearly. Again a table showing the pros and cons of each type of calibration is required otherwise the reader is left figure this out on their own. In fact, it would be good to highlight the different systems in the table.

Page 12 Line 27: The parenthetical note is note needed. I noted later the manuscript mentions a single channel lidar which was confusing as it was noted that there are two channels earlier. This clearly points to the need to be clear on the systems discussed and provide their specific details. Page 13 Lines 1-2: This is obvious and goes to the point noted above. The motivation needs to be first and then the detailed description. It seems that this full description of the polarization analysis is the result of the fact that there are limitations in some of the EARLINET system designs that needed to be addressed. It would be best to state these limitation and how they are addressed. Page 13 Lines 10-14: If I understand this correctly, if one knows the gains using the full gain calibration and the one without then the diattenuation can be determined. Why does one need to use the diattenuation if the full system gain is known? Then it is stated

that the depolarization ratio can be determined using the combination of these gains. Seems that the single gain calibration would be best. Is this method of determining the gain of the upstream optics done less frequent and then applied to the datasets? If so, this is what needs to be stated in the manuscript to understand why this important. It seems to be an after the fact calibration assuming the optics have not changed. Assessment of and the correction for the laser rotation alpha Page 15 Line 4: This is the first time that a single channel polarization lidar has been mentioned. It was assumed from the beginning that there are two channels in the lidars being discussed. Suggest getting rid of this statement or make it clear why a single channel needs to be included. The discussion of the rotation angle is confusing. First, it is not clear if we are talking about setting the rotation angle or correcting datasets using an estimate of the calibration errors. There are many systems and options that are discussed in this section similar to the previous one. It is not clear which systems are capable of different options to change the angle. Later discussions of multiple wavelengths are considered and again there are different options. Suggest that this whole discussion needs to be made more succinct. It is very hard to follow all the systems, wavelengths, and calibration techniques. It would be hard for even someone involved with the systems more directly to follow this discussion. Again, tables with clear definitions and labels would make this better.

There needs to be a clear motivation for the need of the simulations. As noted above, the offset angle of the laser polarization could readily set this within a few degrees just by minimizing the cross polarization channel. This is pretty sensitive if one has a relatively clear air region. Why would there be 10 degree offset as noted in the example provided in section 3.4.3? The same goes for the subsection in this section (correction of alpha in front of the the PBS). As referenced in the introduction and outlined in Alvarez et al. 2006 one can minimize signal ratio in a constant (homogenous) atmosphere quite simply.

The gain ratio eta* The table show deviations of the calibration values with no mention

of the meaning of them. These need to be clearly explained in the paper.

The diattenuation parameter Do The figure shows some examples of the calibration and there is clearly high diattenuation values in some of the systems but why is this important. It appears that this has been not taken into account in previous calibrations, and if this is the case this needs to be noted in the manuscript as the motivation.

Rotation of the plane of polarization of the laser (alpha) Figure 8: Please define the value elastic range-corrected time series. Is this the range squared parallel signal normalized at some altitude? The uncorrected alpha value for these plots needs to be provided. This seems like an extreme case. If so then this needs to be noted. In this case the errors before calibration have no meaning. The improvement in the datasets with the new calibration methods needs to be provided. This gets to the point of the overall conclusions in this paper, what is the improvement to the datasets by going through the calibration methods described. This seems to be the real value of this paper to the data user community to show that there were calibration errors and either corrections are being made or the new methods will be implemented with a stated accuracy.

The discussion of the errors is not clear. The errors needed to be described such that one can know the difference between the standard calibration and also for the various methods. The error assessment needs to be included to derive which one could be considered better. Reasons to not include the retardation errors needs to be provided.

Selected cases of calibrated profiles in the EARLINET framework The discussion on the aerosol types are somewhat speculative and they are not necessary or add significant value to this manuscript. Stating that they are aerosols and likely dust would be sufficient to show that there is enhanced depolarization.

This section has a lot of cases showing the calibrated depolarization from the various systems. The end conclusions of showing these plots is not obvious. A full assessment of the depolarization errors would be better for this manuscript. This requires more justification. For example, the value of the depolarization in the realtively clean regions (i.e. what is the minimum value of the depolarization) gives an indication of the rotation angle and or the ellipticity of the system. This seems to be quite high for system in figure 11 (value of 5) which I am assuming is 5% rather than a ratio of 5. It would be good to state the value of lowest depolarization values in the systems. It is hard to see the magnitude of the scattering in these plots since they are just the signals and not a derived normalized product. There is no scale provided to understand the level of scattering. Provide the backscatter profiles since they were required for the aerosol depolarization results. Evaluation of the gain is more challenging, but an assessment of this parameter would be required to make a conclusion on which technique is better. As stated above, the rotation angle or ellipticity can be evaluated at the low scattering values but it is more challenging to evaluate the gain which requires the high depolarization values. As attempted in the paper, the values can be assessed to some level by looking at ice clouds or dust and provide relative assessments to previous measurements. This does not put an assessment of the accuracy on the data products however. Correlative measurements from multiple systems would be an advancement.

Overall there is not much gained from these plots to really assess the performance of the systems. This really requires a more systematic assessment of the errors looking at specific cases (clear air), calibration stability, maybe one could assess the errors from clouds but this is problematic unless the backscatter ratio can be assessed accurately and there is a clear understanding the cloud microphysical properties which is challenging.

---

## Short Comment (SC1) · 28 Mar 2016

Short Comment on the manuscript by L. Belegante et al. on "Experimental assessment of the lidar polarizing sensitivity"

Let me please thank the AMT-journal for giving me the opportunity to participate to the open discussion on the manuscript proposed by Belegante et al.. This manuscript deals with the important concern of performing accurate polarization lidar measurements. I have several questions on this manuscript, which I hope may benefit to the authors and to potential readers. My comments are in line with the 2nd reviewer and I here bring additional details on what this reviewer states in his report. I also bring a few additional questions.

1. In their manuscript, Belegante et al. discuss on the influence of the lidar optics on

depolarization products and on the effect of the rotation of the plane of polarization of the laser with respect to the PBS. In 2012, I published a paper (G. David et al., Appl. Phys B, 108, 197-216, 2012) which was precisely addressing the same issues through laboratory and field experiments. Could the authors situate their work in regards to other published works and specifically identify what is here new? I feel quite surprised that this paper is not even quoted.

2. In Section 2, the authors use the Stokes-Mueller matrix formalism in a lidar measurement. The Mueller matrix of the emitter optics is there introduced. I introduced how to use the Stokes-Mueller matrix formalism with a pulsed laser source in the specific lidar backscattering geometry in (G. David et al., Opt. Exp., 21, 16, 18624-18639, 2013). There, with my co-authors, we introduced the Mueller matrix of the emitter optics, as it is done here. Could the authors situate their work in regards to this published work? I feel quite surprised, especially in a high ranking journal as AMT.

3. At page 3, the authors speak about "new techniques for performing the calibration of lidar depolarization channels". Could the authors precisely indicate what is intended by new calibration techniques? Moreover, the calibration procedure at +/-45°, which relies on only two points, does not appear, according to the literature, as the most accurate one. Indeed, following Alvarez et al., JTECH, 23, 2006 (from Winker's group, a paper that is quoted) or our paper (G. David et al. 2012), an accurate calibration can be achieved by studying the evolution of the measured depolarization as a function of the angle between the transmitter and the receiver axes. This calibration procedure leads to more accuracy as it relies on several points (around 12), instead of two points as performed in the +/-45° procedure. Even the offset angle can in this way be corrected for, as discussed by Alvarez et al.. We extended his methodology to the case where the HWP is inserted at the emitter in David et al. 2012. Moreover, when using their +/-45° procedure, the authors have to account for possible saturation of the detector. How is it done here? This sounds as a very important issue so as to be quantitative. What is the final assessment of the accuracy of your measurements?

4. As a reader, I found the manuscript very long. Could it be reduced and focused on what is here new ? It would help potential readers.

5. What do the authors intend by "polarizing sensitivity " as proposed in the title of the manuscript? As well-known, the question of sensitivity can be very different from that of accuracy. To perform a sensitivity study, depolarizations less than the percent range should be addressed, as we discussed in G. David et al., in 2012.

6. Many lidar stations are nowadays equipped with dual-wavelength capabilities. How do you account for the influence of wavelength-dependent optics in your study ? What are the wavelength cross-talks and their influence on the polarization retrievals? This issue sounds important in a manuscript where the authors deal with "correcting the influence of the receiver optics". I recall that a prerequisite to the polarization lidar calibration is that the detector transfer matrix be diagonal with negligible wavelength and polarization cross-talks, as we published (David et al. 2012). If not, this may lead to quite substantial errors in the measured depolarization ratio.

---

## Editor Comment (EC1) · A. Ansmann (Editor) · 1 Apr 2016

Dear authers!

Please withdraw your paper!

Reviewer #2 votes for rejection and voted POOR regarding the Presenation Quality.

So, there is no alternative for me to reject the paper.

Please consider all three reviewer comments carefully and re-submit your paper as soon as possible.

The dead line for contributions to of the Special Issue now is 31 October 2016